

# An update on anticancer drug development and delivery targeting carbonic anhydrase IX

Justina Kazokaitė[1,*], Ashok Aspatwar[2,3,*], Seppo Parkkila[2,3] and Daumantas Matulis[1]

[1] Department of Biothermodynamics and Drug Design, Institute of Biotechnology, Vilnius University, Vilnius, Lithuania
[2] Faculty of Medicine and Life sciences, University of Tampere, Tampere, Finland
[3] Fimlab Ltd, Tampere, Finland
[*] These authors contributed equally to this work.

## ABSTRACT

The expression of carbonic anhydrase (CA) IX is up-regulated in many types of solid tumors in humans under hypoxic and acidic microenvironment. Inhibition of CA IX enzymatic activity with selective inhibitors, antibodies or labeled probes has been shown to reverse the acidic environment of solid tumors and reduce the tumor growth establishing the significant role of CA IX in tumorigenesis. Thus, the development of potent antitumor drugs targeting CA IX with minimal toxic effects is important for the target-specific tumor therapy. Recently, several promising antitumor agents against CA IX have been developed to treat certain types of cancers in combination with radiation and chemotherapy. Here we review the inhibition of CA IX by small molecule compounds and monoclonal antibodies. The methods of enzymatic assays, biophysical methods, animal models including zebrafish and *Xenopus* oocytes, and techniques of diagnostic imaging to detect hypoxic tumors using CA IX-targeted conjugates are discussed with the aim to overview the recent progress related to novel therapeutic agents that target CA IX in hypoxic tumors.

## Introduction

Recent advances in cancer therapy show that hypoxia is the major contributor to tumor development (*Semenza, 2014*; *Hanahan & Weinberg, 2011*). The poor and chaotic tumor angiogenesis leads to the insufficient oxygen and nutrient supply which drastically affects the cellular metabolism (*Welti et al., 2013*). Due to the up-regulated glycolysis, tumor cells produce increased amounts of lactate and protons. As a consequence of mTORC1&2 mediated functional and transcriptional activation of c-Myc, tumor cells tend to metabolize glucose preferably via glycolysis rather than oxidative phosphorylation despite sufficient levels of oxygen. This phenomenon is known as Warburg effect (*Warburg, 1956*; *Vander Heiden, Cantley & Thompson, 2009*). The resultant hypoxic and acidic extracellular milieu

Corresponding author
Daumantas Matulis, matulis@ibt.lt, daumantas.matulis@bti.vu.lt

significantly increases the resistance of cancer cells to chemotherapy and radiotherapy as well as promotes invasiveness and metastasis (*Wojtkowiak et al., 2011*; *Good & Harrington, 2013*).

Hypoxia stimulates crucial pathways, one of which is implemented by the activation of the heterodimeric hypoxia-inducible factor (HIF) (*Denko, 2008*). This hypoxia-induced transcriptional program is important for tumor cells to survive harsh conditions. There are many downstream-target genes of HIF, which encode proteins, such as adhesion molecules (*Ryu et al., 2010*), matrix metalloproteinases (*O'Toole et al., 2008*), chemokine receptors (*Li et al., 2009a*), growth factors (*Kotch et al., 1999*), differentiation proteins (*Takubo et al., 2010*), glycolytic enzymes (*Obach et al., 2004*), lactate transporters (*Ullah, Davies & Halestrap, 2006*), and ion transporters (*Parks, Chiche & Pouysségur, 2013*). Some HIF-regulated proteins have been shown to be hypoxia-related anticancer targets and possess therapeutic applications (*Wilson & Hay, 2011*). Thus, HIF is critically essential for cancer cells to survive and metastasize in the hostile tumor environment due to the HIF-dependent activation of oncogenes and inactivation of tumor suppressor genes.

As a consequence of HIF-mediated transcriptional response to tumor hypoxia, the intracellular and extracellular pH is unbalanced. Normal cells differ from cancer cells by the mechanisms of pH regulation, which create the reversed pH gradient in tumors. Physiologically the intracellular pH ($pH_i$) is lower than the extracellular pH ($pH_e$), which is ~7.4. Pathologically $pH_i$ is higher than $pH_e$, which is 6.7–7.1 (*Hashim et al., 2011*; *Mazzio, Smith & Soliman, 2010*). This phenomenon of extracellular acidification under hypoxic conditions is created by HIF-dependent induction of proteins, such as transmembrane enzymes, ion pumps, and transporters. They export lactate and protons and import bicarbonate ions to optimize the tumor progression. Key pH-regulators are V-ATPase, $Na^+/H^+$ exchanger (NHE), monocarboxylate transporters (MCTs) and carbonic anhydrase (CA) IX.

There are seven evolutionarily distinct CA gene families: α-, β-, γ-, δ-, ζ- η-, and θ-CAs (*Prete et al., 2014*; *Supuran & Capasso, 2015*; *Krishnamurthy et al., 2008*; *Kikutani et al., 2016*; *Aggarwal et al., 2013*; *Capasso & Supuran, 2015*). In humans, there are 15 α-CA isoforms, of which 12 are catalytically active and exhibit diverse enzymatic activity, various cellular distribution and physiological functions (*Frost, 2014*). Being a member of α-CA isoforms in human body, CA IX is a transmembrane homodimer, which catalyzes the reversible hydration of carbon dioxide to bicarbonate and proton outside the cell. The intracellular pH of cancer cells is regulated by the export of lactate and protons and on the import of bicarbonate ions generated by the hydration of $CO_2$. The acidic metabolites accumulate pericellularly because of the ineffective tumor vasculature and extracellular acidosis. To reduce changes of intracellular pH, the bicarbonate is transported into the cell through the bicarbonate transport metabolon composed of CA IX and bicarbonate transporters. Thereby CA IX is important for cancer cell proliferation because of the participation in both processes: the extracellular acidification and the intracellular alkalinization (*Aggarwal et al., 2013*; *Alterio et al., 2009*).

CA IX is relevant not only for the cancer cell survival, but also to several other biological processes, such as the maintenance of cancer stem cell (CSC) function, migration, and

invasion. Cell migration depends on the formation of lamellipodia, which have been shown to be partially produced by activation of CA IX and its interaction with bicarbonate transporters (*Svastova et al., 2012*). In addition, acidosis under hypoxic conditions activates proteolytic enzymes, which degrade the extracellular matrix and promote metastasis formation. Thus, CA IX targeting compounds have shown to significantly diminish the cancer stem cell population, inhibit the growth of primary tumors, and reduce metastatic burden (*Swietach et al., 2010*; *Pastorek & Pastorekova, 2015*; *Sedlakova et al., 2014*; *Lock et al., 2013*; *McDonald et al., 2010*).

In normal tissues, the expression of CA IX is negligible with the exception of the stomach and gallbladder epithelia (*Pastorekova et al., 1997*). There is a broad spectrum of aggressive malignancies, where CA IX is predominantly overexpressed, namely, neuroblastoma (*Ameis et al., 2016*), breast tumor (*Betof et al., 2012*), head and neck tumors (*Yang et al., 2014*), ovarian tumor (*Choschzick et al., 2011*), pancreatic tumor (*Couvelard et al., 2005*), hepatocellular carcinoma (*Huang et al., 2015*), etc. In addition, there are several reviews, which summarize the significance of CA IX as a promising biomarker for the tumor development (*Van Kuijk et al., 2016*). Thus, CA IX has emerged as the clinically relevant biomarker and a potential anticancer-drug target.

At the core of $\alpha$-CA active site, the metal ion, Zn (II), is tetrahedrally coordinated to three imidazole rings from His94, 96, and 119 (numbering according to CA II) and a water/hydroxide anion (*Fisher et al., 2007*). The catalytic site is located at approximately 15 Å depth conical cavity which consists of hydrophobic (Val121, Val143, Leu198, Val207, Trp209) as well as hydrophilic (Tyr7, Asn62, His64, Asn67, Thr199, Thr200) regions and provides the accessibility to the solvent (*Krishnamurthy et al., 2008*; *Eriksson, Jones & Liljas, 1988*; *Pocker & Sarkanen, 1978*).

A high conservation of amino acids in the active site and surrounding faces has been found among the 12 catalytically active human CA isoforms (*Aggarwal et al., 2013*; *Pinard et al., 2015*). Thus, the design of CA isoform-selective inhibitors has been the challenging goal for many researchers. In 1954, acetazolamide was approved in clinic as the first CA-targeting antiglaucoma drug (*Supuran, 2012*). In the next decades, a vast collection of CA inhibitors with various affinities and selectivities has been designed and has been extensively reviewed (*Lomelino & McKenna, 2016*; *Supuran, 2016*; *Supuran, 2017*; *Alterio et al., 2012*; *Monti, Supuran & De Simone, 2013*).

It is a challenging task to design inhibitors that would be not only highly selective to CA IX, but also safe for use in humans for the treatment and diagnosis of hypoxic tumors. Many aspects need to be considered to achieve the final goal of developing the promising drugs, that could selectively inhibit CA IX in hypoxic tumors. The knowledge about the active site structure of the protein and permeability of the inhibitor across the cell membrane is essential for designing the CA IX specific inhibitors. An inhibitor may be selective for CA IX, but it may need to be attached to a conjugate to make it impermeable through the membrane.

Similarly, the potential inhibitors need to go through the physical and biochemical screening and various modifications to develop as CA IX isoform specific compounds. The most promising CA IX inhibitors have to be screened for safety and toxicity *in vivo* using

animal models, such as zebrafish, before subjecting them to preclinical characterization. In addition to chemical compounds, CA IX-selective biological molecules, such as monoclonal antibodies (mAbs), are at various stages of preclinical and clinical trials as potential anticancer agents targeting CA IX in hypoxic tumors. In addition, the anticancer agents based on CA IX selective inhibitors can be conjugated with various probes for the diagnosis of hypoxic tumors.

## SURVEY METHODOLOGY

A wide variety of chemical compounds have been described in the literature that target tumor-associated CA IX. In this review, we selectively describe only aromatic sulfonamides that have been demonstrated to bind and inhibit the catalytic domain of recombinant human CA IX by at least two experimental approaches, such as inhibition of enzymatic activity and biophysical assays including the fluorescent thermal shift assay (FTSA), isothermal titration calorimetry (ITC), and surface plasmon resonance (SPR). We emphasize the use of non-mammalian animal models, such as zebrafish and *Xenopus* oocytes for the toxicity, affinity, and selectivity studies of CA IX targeting sulfonamides. Published in 2016–2017, these studies suggest possibilities that could help in the development of antitumor agents prior to preclinical characterization in mice models.

For reviewing the information, we identified the articles containing information about different biological and chemical antitumor agents that target CA IX in hypoxic tumors. The literature search was performed using the relevant keywords in PubMed. For example, the antibody section was compiled with all available articles published since 1986 up to 2017, in which the use of antibodies for the detection of CA IX in patients was described. Publications were retained if they contained relevant information about the promising agents that target CA IX in humans and also during the development of these agents in human cell lines and mice models. Priority was given to the antitumor agents that have been developed either for the treatment or imaging of the tumors using novel strategies.

The focus of this review is also to present recent developments in the treatment and diagnosis of solid tumors under hypoxic conditions that express CA IX. We present the recent achievements on the 8 diagnostic tools including chemical and biological antitumor agents targeting CA IX that are at various stages of preclinical and clinical trials for treating the hypoxic tumors. This review combines the information about animal models, enzymatic, biophysical methods used in CA field, as summarized in Fig. 1, with the latest references of novel anticancer agents that are currently applied to target CA IX for the diagnosis and treatment.

## CA INHIBITOR ASSAYS

### CA enzymatic activity inhibition assay

To evaluate the potency of CA-targeting inhibitor, the stopped-flow $CO_2$ hydration assay (SFA) has been widely applied for more than five decades since the discovery of the method to measure CA catalyzed $CO_2$ hydration rate by Gibbsons and Edsall and by Khalifah (*Gibbons & Edsall, 1963*; *Gibbons & Edsall, 1964*; *Khalifah, 1971*). This approach

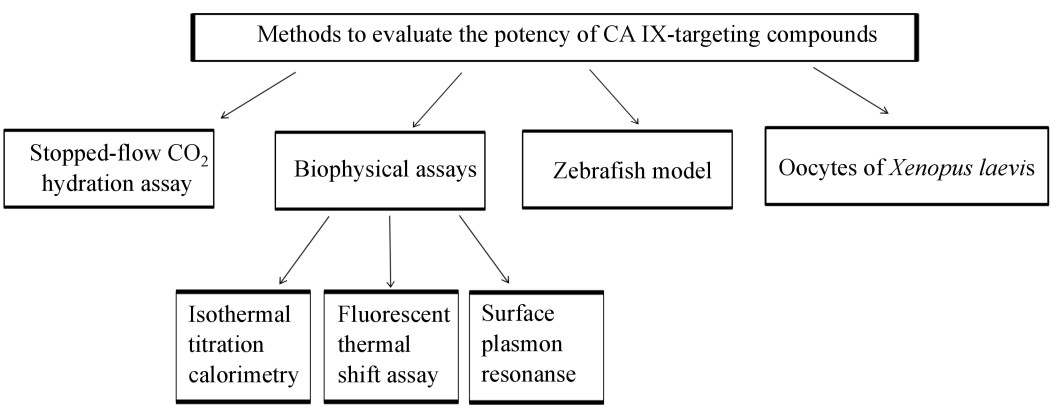

**Figure 1** **Methods which might be applied for developing CA IX-targeting compounds before pre-clinical characterization in tumor cells and mice.**

is based on the monitoring of the changes in absorbance of pH sensitive indicator upon CA catalyzed $CO_2$ hydration reaction. The half-maximal inhibitory concentration, $IC_{50}$, is determined by fitting the compound dose curve according to the Hill model or Morrison equation (*Morrison, 1969*). The inhibition constant, $K_i$, can be obtained from $IC_{50}$ value by Cheng-Prusoff equation (*Cheng & Prusoff, 1973*).

Supuran and co-authors have developed a large library of CA inhibitors by SFA and divided them into five groups according to CA inhibition mechanisms: (1) the zinc binders (sulfonamides and their isosteres, dithiocarbamates and their isosteres, hydroxamates, etc.) (*Supuran, 2012*; *Alterio et al., 2012*; *Carta et al., 2013*; *Innocenti, Scozzafava & Supuran, 2010*; *Carta et al., 2012*; *Supuran, 2013*); (2) compounds that anchor to the zinc-coordinated water molecule/hydroxide ion (phenols, polyamines, sulfocoumarins, etc.) (*Nocentini et al., 2016*; *Davis et al., 2014*; *Carta et al., 2010*; *Innocenti et al., 2008*; *Santos et al., 2007*); (3) inhibitors which occlude the entrance to the CA active site (coumarins and their isosteres) (*Nocentini et al., 2015*; *Bozdag et al., 2017*; *Tars et al., 2013*); (4) compounds which bind out of the active site (carboxylic acid derivates) (*D'Ambrosio et al., 2015*); (5) inhibitors which bind in an unknown way (secondary/tertiary sulfonamides, imatinib, nilotinib, etc.) (*Parkkila et al., 2009*; *Supuran, 2016*; *Métayer et al., 2013*). Since these various compounds have been subject of numerous recent reviews, here we concentrate only on aromatic sulfonamides as CA inhibitors. Supuran's group also measured the affinity of monoclonal antibodies to target CA isoforms using SFA (*Dekaminavičiūtė et al., 2014*). In addition to other previously synthesized compounds containing fluorine, our group has identified a series of fluorinated benzenesulfonamides as strong CA IX inhibitors by SFA and have shown a correlation between parameters obtained by enzymatic and biophysical assays (*Dudutienė et al., 2014*).

Importantly, CA isoforms share not only hydratase, but also esterase activity which was discovered in early 1960s (*Tashian, Douglas & Yu, 1964*). Both reactions occur in the same catalytic pocket suggesting similarities in their mechanisms. The method to determine

esterase activity is a high-throughput colorimetric assay with various applications, such as screening chemical molecules or antibodies against CA isozymes (*Akıncıoğlu et al., 2013*; *Uda et al., 2015*).

## Biophysical assays of inhibitor binding to CAs

Advantages and limitations of enzymatic inhibition versus biophysical assays of inhibitor binding have been assessed and are compared in our recent manuscript (*Smirnovienė, Smirnovas & Matulis, 2017*). Biophysical methods not only determine the thermodynamic parameters of ligand binding to CAs, but also provide insight into numerous significant factors, which influence the binding: local water structure, hydrogen bonding, hydrophobic interactions, and desolvation. The thermodynamic profiles of drug candidate binding to CA have been widely used. Here we will focus on biophysical techniques, such as fluorescent thermal shift assay (FTSA), isothermal titration calorimetry (ITC), and surface plasmon resonance (SPR), which have been applied in the rational drug design of isoform-selective CA inhibitors.

### Isothermal titration calorimetry

Since the invention of first analog of an isothermal titration calorimeter in 1966 (*Izatt et al., 1966*; *Christensen et al., 1966*) and its modifications for biological applications in 1980s (*Ramsay, Prabhu & Freire, 1986*; *Schön & Freire, 1989*), ITC has become the method of choice to study protein target-ligand interactions. During the experiment, in the current commercial titration calorimeters, the inhibitor solution from the syringe is injected at constant temperature into the protein solution preloaded to the calorimeter cell until all binding sites of the protein become occupied by the ligand. Importantly, ITC does not require the inhibitor or protein to be labeled or immobilized and allows the determination of the affinity, the binding enthalpy and the stoichiometry in a single titration experiment (*Klebe, 2015*; *Krimmer & Klebe, 2015*; *Geschwindner, Ulander & Johansson, 2015*; *Falconer, 2016*).

Numerous studies of interactions between diverse ligands and target CA isoforms have been performed by ITC (*Krishnamurthy et al., 2008*; *DiTusa et al., 2001*; *Khalifah et al., 1993*). The binding of anions to CA II was evaluated using ITC, X-ray crystallography, and molecular dynamics simulations by Whitesides group (*Fox et al., 2015*). For the deeper understanding of structure–activity relationships, the analysis of buffer ionization effects was performed by ITC upon an inhibitor binding to recombinant human CA isoforms, including CA I (*Morkūnaitė et al., 2015*), CA II (*Morkūnaitė et al., 2015*), CA VB (*Kasiliauskaitė et al., 2015*), CA VI (*Kazokaitė et al., 2015*), CA VII (*Pilipuitytė & Matulis, 2015*), CA IX (*Linkuvienė et al., 2016*), CA XII (*Jogaitė et al., 2013*), and CA XIII (*Baranauskienė & Matulis, 2012*). In addition, ITC standard and displacement titrations were combined with the X-ray crystallographic structures to determine the intrinsic, buffer-independent affinity of *para* substituted tetrafluorobenzenesulfonamides binding to several human CA isoforms (*Zubrienė et al., 2015*).

### Fluorescent thermal shift assay

FTSA, also called differential scanning fluorimetry and, in high-throughput format, ThermoFluor®, has been widely applied by numerous researchers and companies, such as

Johnson & Johnson, New Brunswick, United States. It is a rapid screening method in the drug discovery to measure the binding affinities of chemical compounds to targets (*Kranz & Schalk-Hihi, 2011*; *Lo et al., 2004*; *Pantoliano et al., 2001*; *Niesen, Berglund & Vedadi, 2007*). FTSA monitors the equilibrium of a protein between its folded and unfolded states by detecting the fluorescence of solvatochromic probes, such as 1,8-anilinonaphthalene sulfonate or SYPRO® orange, while the temperature is steadily increased. This method determines the protein melting temperature which can be highly affected by the affinity of ligand and its concentration (*Cimmperman & Matulis, 2011*; *Cimmperman et al., 2008*). In addition, FTSA is a convenient technique to characterize protein thermal stabilities at various conditions including diverse buffers, excipients, etc (*Mezzasalma et al., 2007*; *Cummings, Farnum & Nelen, 2006*).

FTSA has been widely applied in the search of CA inhibitors. The binding of sulfamate and sulfamide derivatives to human CA II was investigated using FTSA by *Klinger et al. (2006)*. FTSA was also applied by our group to investigate the interactions between human CA isoforms and various series of inhibitors, including tri- and tetrafluorobenzenesulfonamides (*Dudutienė et al., 2013*; *Dudutienė et al., 2015*), benzenesulfonamide derivatives with pyrimidine moieties (*Čapkauskaitė et al., 2013*), saccharin sulfonamides (*Morkūnaitė et al., 2014*), benzenesulfonamides with benzimidazole moieties (*Zubrienė et al., 2014*), 4-amino-substituted benzenesulfonamides (*Rutkauskas et al., 2014*). In addition, the profiles of thermal stabilities of recombinant human CA VB (*Kasiliauskaitė et al., 2015*), CA VI (*Kazokaitė et al., 2015*), CA IX (*Linkuvienė et al., 2016*), and CA XII (*Jogaitė et al., 2013*) was described using FTSA.

### Surface plasmon resonance

SPR was first demonstrated for the monitoring of biomolecular interactions by Lundstrom et al. in 1983 (*Liedberg, Nylander & Lunström, 1983*) and the first commercial SPR instrument was launched by Pharmacia Biosensors AB in 1991 (*Jönsson et al., 1991*). During the last decades, SPR biosensors have become the state-of-the-art technology in diagnostics and biomedical research to determine a real-time kinetics and binding affinities of ligand-protein interactions. To screen lead compounds, one of the binding partners, usually the target protein, is immobilized on a metal surface and the ligand flows over that surface by microfluidic system. SPR is a label-free optical method, which measures the changes in refractive index at the metal surface upon the binding reaction.

Studies of SPR application in CA research used recombinant human CA I (*Jecklin et al., 2009*) or mostly CA II (*Myszka, 2004*; *Navratilova & Hopkins, 2010*; *Papalia et al., 2006*) isoform as a model for the screening of numerous inhibitors. In contrast, Talibov et al. immobilized six human recombinant CA isoforms (full-length CA I, CA II, CA VII, CA XIII, catalytic domains of CA IX and CA XII) and analyzed their interactions with 17 benzenesulfonamide ligands by SPR. Interestingly, results revealed one compound from investigated series to be as a tight binder to recombinant CA IX with the dissociation rates too slow to be determined by SPR (*Talibov et al., 2016*).

## Zebrafish model for compound toxicity

Phenotype-based screening using zebrafish has become a promising high-throughput assay for the drug discovery. This approach revealed that 62%, of drugs approved from 1999 till 2008, were discovered by phenotype-based screens despite that they represented only a small fraction of all screens (*MacRae & Peterson, 2015*). Phenotypic screens possess many significant advantages over target-based screens including the identification of drugs without a validated target or the characterization of the therapeutic profile of the compound, which affects several targets simultaneously. Zebrafish has emerged as a powerful model system for phenotypic screens of drug-candidates *in vivo* because of many advantages that include high homology between zebrafish and mammalian CAs, low cost, and avoidance of most ethical issues associated with the use of other animals. However, zebrafish lack lung, prostate, and mammary glands, heart septation, limbs, and it is necessary to grow zebrafish at 30 °C, while compounds against mammalian targets are usually optimized for 37 °C (*Lin, Chiang & Tsai, 2016*; *Rennekamp & Peterson, 2015*).

Zebrafish can be particularly useful to carry out toxicological studies of CA inhibitors. Toxic effects of two fluorinated benzenesulfonamides as CA IX inhibitors were investigated on zebrafish development (*Kazokaitė et al., 2016b*). $LC_{50}$ values showed that one compound exhibited 10-fold lower toxicity than ethoxzolamide (EZA), a compound used as a drug in humans. In addition, light-field microscopy and histological analysis revealed that EZA induced side effects such as pericardial edema, unutilized yolk sac and abnormal body shape of zebrafish. In contrast, developmental abnormalities were not detected in embryos treated with the fluorinated benzenesulfonamides (Table 1). Thus, this study showed that CA IX inhibitors did not have adverse effects on phenotype and morphology of zebrafish larvae. Such toxicological screenings of the compounds using zebrafish could provide information on the safety of lead molecule that could be useful for further development into a drug.

## Oocyte system for heterologous expression of CAs to determine compound affinity and selectivity

Since 1960s, the *Xenopus laevis* has been widely used as a convenient animal model in various biomedical fields including molecular and physiological research. The *Xenopus* oocytes have many advantages including a large number of offspring, easy manipulations because of their big size (1.1–1.3 mm) and easy maintenance. Furthermore, oocytes feature highly efficient translation of heterologous RNA into protein.

Native *Xenopus* oocytes do not possess any CA activity and thus have become a convenient *in vivo* model system to investigate CA inhibitors. The enzymatic activity of CA can be evaluated with microelectrodes while monitoring the intracellular and extracellular acidification. Results can be confirmed by mass spectrometric gas analysis of lysed or intact oocytes (*Becker, 2014*). The transfection of *Xenopus* oocytes with cRNA of CA isozymes has been published by Deitmer's group (*Klier et al., 2016*; *Schneider et al., 2013*). They showed the complete inhibition of CA IX enzymatic activity with 30 μM EZA according to the rates of cytosolic pH changes and amplitudes of pH changes at the outer membrane side (*Klier et al., 2016*). The same effect was found in CA IX expressing

**Table 1  Biological model systems for the investigation of CA IX inhibitors.** The compounds did not show any significant toxicity on zebrafish and possessed nanomolar $IC_{50}$ for heterologous CA IX expressed in *Xenopus* oocytes. In addition, the selectivity of compounds toward CA isoforms was evaluated according to the effect of compounds on the reduction of extracellular (CA IX, CA IV, and CA XII) and intracellular (CA II) CA-induced acidification in oocytes (*Kazokaitė et al., 2016a*; *Kazokaitė et al., 2016b*).

| Inhibitor / Type of study | VD11-4-2 | VD12-09 |
|---|---|---|
| |  |  |
| **Toxicology**  | $LC_{50} = 120\ \mu M$ | $LC_{50} = 13\ \mu M$ |
| | *Methods*: 1. light-field microscopy  2. histological analysis | |
| **Affinity and selectivity**  | CA IX: $IC_{50} = 25$ nM  CA II: <5.0% effect on pH at 10 µM  CA IV: 57.8% effect on pH at 10 µM  CA XII: 28.0% effect on pH at 50 nM | CA IX: 25.5% effect on pH at 10 µM  CA II: <5.0% effect on pH at 10 µM |
| | *Methods*: 1. pH monitoring with microelectrodes  2. mass spectrometric gas analysis | |

$IC_{50}$ - the concentration causing 50% inhibition of target activity, $LC_{50}$ - 50% lethal concentration.

oocytes treated with 1 µM fluorinated benzenesulfonamide targeting CA IX (*Kazokaitė et al., 2016a*). The $IC_{50}$ was found to be in the range of 15–25 nM for both intracellularly and extracellularly expressed CA IX. Moreover, the compound exhibited strong selectivity over CA II, CA IV or CA XII in oocytes expressing a particular CA isoform (Table 1). This novel *in vivo* approach allows the identification of the affinity and selectivity of CA IX inhibitors in the living eukaryotic cell with fully matured target CA isozyme.

## CA IX-TARGETED STRATEGIES

Targeting CA IX enzyme is a promising approach for the development of new therapeutics against hypoxic tumors. There are several agents that can selectively target CA IX by using different strategies. Here, we present therapeutic agents that have been used against CA IX for diagnosis and treatment of hypoxic tumors in humans (Table 2).

### Monoclonal antibodies for CA IX-targeted therapy

M75 and chimeric G250 (cG250) are two widely-applied monoclonal antibodies (mAbs) recognizing human CA IX. These mAbs have been used for clinical detection or therapy (*Oosterwijk et al., 1986*; *Závada et al., 1993*). The M75 targets the PG-domain of CA IX and is used for the detection of CA IX in human tissues (*Chrastina, Pastoreková & Pastorek, 2003*; *Chrastina et al., 2003*; *Zatovicova et al., 2010*). cG250 has been successfully developed

**Table 2  Anti-tumor agents for targeting hypoxia-induced CA IX for therapy and diagnosis.**

| Anti-tumor agents | Therapy stage | Diagnosis | References |
|---|---|---|---|
| SLC-0111 | Phase I trial | Solid tumors | *Welichem Biotech Inc & Ozmosis Research Inc (2014)* |
| U-104 | Preclinical trials | Xenograft tumor model (pancreatic ductal adenocarcinoma cell line Pt45.P1/asTF+) | *Pacchiano et al. (2011)*, *Lou et al. (2011)*, *Ramchandani et al. (2016)* |
| G250 (girentuximab) | Phase III clinical trial | ccRCC diagnosis | *Wilex (2004)* |
| $^{177}$Lu-labelled girentuximab | Phase II clinical trials | ccRCC diagnosis | *Pal & Agarwal (2016)* |
| Indisulam | Phase I clinical trials | Solid tumors | *Dittrich et al. (2007)*, *Eisai Limited (2005)* |
| NIR fluorescent derivative of the acetazolamide | Preclinical trials | Xenograft tumor model | *Tafreshi et al. (2012)* |
| $^{99m}$Tc-(HE)3-ZCAIX:2 | Preclinical trials | Disseminated cancer | *Garousi et al. (2016)* |
| $^{125}$I-ZCAIX:4 | Preclinical trials | Primary renal cell carcinoma | *Garousi et al. (2016)* |

for anticancer immunotherapy (*Cardone, Casavola & Reshkin, 2005*) due to its ability to elicit antibody-dependent cellular cytotoxicity (*Surfus et al., 1996*). The clinical trials showed that cG250 is safe, and has effect on the disease burden, when applied alone or together with interferon-α (*Davis et al., 2007*; *Siebels et al., 2011*). This mAb is marketed by WILEX AG using RENCAREX® as a trade name and has been used for renal cell carcinoma patients (RCC) who are at high risk of relapse (*McDonald et al., 2012*). In the recent past, this mAb under the name of girentuximab, has been assessed as an adjuvant in Phase III ARISER trial in RCC patients and showed that the patients expressing CA IX benefited more than ones without or minimal expression of CA IX (*Wilex, 2004*). In a phase II study, the mAb labeled with lutetium ($^{177}$Lu-girentuximab) demonstrated the significantly positive impact on the progressive metastatic ccRCC patients (*Pal & Agarwal, 2016*). In addition, REDECTANE® ($^{124}$I-girentuximab) has been in clinical development targeting ccRCC (*Wilex, 2017*). Furthermore, A3 and CC7 have been developed as CA IX-selective mAbs by the phage display method. They showed promising results in animal models of colorectal cancer and may be useful for the drug delivery (*Oosterwijk et al., 1986*). These studies clearly showed that mAbs and their modified versions are potential candidates for the development as anticancer agents targeting tumors that express CA IX.

Several monoclonal antibodies have been developed that influence the catalytic activity of CA IX (*Zaťovicová et al., 2003*). Pastorekova's group has demonstrated that the mAb VII/20 binds to the catalytic domain of CA IX, causing the receptor-mediated internalization of the antibody-protein complex. Authors have shown that this process is important for the immunotherapy because significant anticancer effects of VII/20 were found in mouse xenograft model of colorectal carcinoma (*Zatovicova et al., 2010*). Thus, the application of CA IX-targeting antibodies might be significantly beneficial immunotherapeutic strategy.

Furthermore, mAbs have been considered as the ligands of choice for the design of antibody-drug conjugates (ADCs). In current clinical development, there are 65 ADCs mostly targeting various proteins at cell surface (*Xu, 2015*; *Beck et al., 2017*). Since antibodies might cause problems related with the penetration or immunogenicity, there is a demand for smaller agents, such as peptides or chemical derivatives, for the drug delivery. Recently, Neri with co-authors has described CA IX-targeting small-molecule drug conjugates. Monovalent and divalent conjugates of acetazolamide with the cytotoxic maytansinoid DM1 exhibited promising anticancer activity in SKRC52 renal cell carcinoma *in vivo* (*Krall, Pretto & Neri, 2014*; *Krall et al., 2014*).

## Chemical compounds targeting CA IX for therapy

A wide range of CA IX selective inhibitors has been designed with the help of X-ray crystallography and computational analysis. Among them, a group of sulfonamides show potential for developing as anticancer agents. A sulfonamide compound, indisulam, has shown a significant antitumor activity in preclinical cancer models (*Dittrich et al., 2007*). Phase II clinical trials were conducted to determine the efficacy, safety and tolerability of indisulam in combination with irinotecan in patients with metastatic colorectal cancer who were previously treated with 5-fluorouracil/leucovorin and oxaliplatin (*Eisai Limited, 2005*) but no further information is available about the outcome of the trial. Similarly, bis-sulfonamides have shown promising results *in vitro* in tumor sections and target tumors *in vivo* (*Buller et al., 2011*). Preclinical studies using ureidosulfonamide inhibitor of CA IX, named as U-104 or SLC-0111 (SignalChem Lifesciences Corp, Richmond, BC, CA), showed positive effects with the negligible toxicity for the treatment of various tumors (*Pacchiano et al., 2011*; *Lou et al., 2011*). Recently, U-104 has been demonstrated to be effective *in vitro* and *in vivo* models of the pancreatic ductal adenocarcinoma (Pt45.P1/asTF +). U-104 significantly decreased the growth of pancreatic cells in hypoxia but not in normoxia and reduced the tumor growth in mice emphasizing the potential of the compound as a therapeutic agent against CA IX (*Ramchandani et al., 2016*).

Small molecule-drug conjugates (SMDCs) have been used for the selective delivery of therapeutic agents to tumor sites. The series of stable and therapeutically active SMDCs were generated by attaching acetazolamide to monomethyl auristatin E using dipeptide linkers. They showed a promising antitumor activity in mice bearing SKRC-52 renal tumors. Since CA IX is a transmembrane protein, the findings of this study is significantly important for the targeted drug delivery in kidney cancer patients (*Corso & Neri, 2017*). Similarly, PEGylated bis-sulfonamide CA inhibitors were synthesized from aminosulfonamide pharmacophores conjugated with either ethyleneglycol oligomeric or polymeric diamines. These compounds efficiently controlled the growth of several CA IX-expressing cancer cell lines including colon HT-29, breast MDA-MB-23, and ovarian SKOV-3 (*Akocak et al., 2016*).

To demonstrate the antitumor effect of CA IX inhibition *in vivo*, the vast library of conjugates against CA IX has been designed. Dual targeting bioreductive nitroimidazole-based sulfamide drug, named as DH348, was used to evaluate the impact on the extracellular acidification and radiosensitivity in HT-29 colorectal cancer cells and mouse xenograft

models. By using nontoxic doses of DH348, the hypoxia-induced extracellular acidification was significantly reduced and the tumor growth was decreased. DH348 also sensitized the tumor to irradiation and the effect was CA IX-dependent (*Dubois et al., 2013*). In addition, the combination of SLC-0111 and APX3330 has been reported in patient-derived 3D pancreatic cancer models. Results of dual treatment showed a greater decrease in the intracellular pH and 3D tumor spheroid growth than treatment with either inhibitor alone (*Logsdon et al., 2016*). Recently, phase I clinical trial of SLC-0111 has been finished and the compound was scheduled to enter Phase II trials (*Welichem Biotech Inc & Ozmosis Research Inc, 2014*). Since results of phase I trials have not been published, the characterization of pharmacodynamics and pharmacokinetics of SLC-0111 is not available yet.

### Targeting CA IX using nanoparticles

Gold nanoparticles coated with chemical inhibitors are a relatively new to the field of the development of agents targeting CA IX. The gold nanoparticles modified with CA IX inhibitors cannot pass through the membrane. Thus, they show a great potency to be effective in targeting and inhibiting the extracellular active site of CA IX.

The nanoparticles, which were modified with thiols and benzenesulfonamide groups, selectively inhibited CA IX ($K_i$ 32 nM) but their affinities toward CA I and CA II were more than 10-folds lower ($K_i$ 451 nM). In addition, these nanoparticles possessed a greater affinity toward CA IX than acetazolamide and may be suitable candidates for imaging and treatment of hypoxic tumors (*Stiti et al., 2008*). Recently, gold nanoparticles were used to target CA IX for photoacoustic imaging and optical hyperthermia (*Supuran & Winum, 2015*). In addition, derivatives of benzenesulfonamides combined with nanorods showed a significant impact on the reduction of the extracellular acidification in hypoxic human mammary and colorectal carcinomas (*Ratto et al., 2015*). These studies suggest that the use of nanoparticles can be used to efficiently target extracellular part of CA IX in hypoxic tumors.

To improve the potency and selectivity of novel inhibitors, recently multivalent nanoconstructs have been developed (*Touisni et al., 2015*; *Kanfar et al., 2015*). These nanoconstructs showed excellent inhibitory effects with $K_i$ values of 6.2–0.67 nM against tested CA isozymes. They contain multiple copies of a ligand, which are displayed closely on the same derivative. Thus, a weak mM binder can be changed to nM binder and the biomolecular recognition can be enhanced (*Kanfar et al., 2017*). Even though the use of multivalent nanoconstructs in the field of CA IX inhibition is quite recent, there is a great potential to develop CA IX inhibitors with high affinity and selectivity properties using this multivalent strategy.

## IMAGING METHODS

Detection of hypoxic regions of solid tumors is an important step for cancer treatment (*Bertout, Patel & Simon, 2008*). The application of selective ligands against CA IX in diagnostic imaging has been widely investigated. They could help to decide which patients can benefit from the adjunctive therapy (*Höckel et al., 1996*). Both antibodies and

small molecular weight compounds have been used for non-invasive imaging of CA IX in a number of aggressive and late stage types of tumors and metastases (*McDonald et al., 2012*).

## Imaging of tumors using CA IX-specific mAbs

CA IX is a useful biomarker for clear cell renal cell carcinoma (ccRCC) because CA IX is absent in normal kidney tissues. The CA IX-specific cG250, radiolabeled with iodine-124 or zirconium-89, has been used for the diagnosis of ccRCC (*Stillebroer et al., 2007*). High parameters of sensitivity and specificity were determined by positron emission tomography/computed tomography (PET/CT) when cG250 labeled with iodine-124 was applied for the imaging of ccRCC (*Divgi et al., 2007*). This study suggests a great potential to monitor ccRCC in patients and allows the differentiation of ccRCC versus non-ccRCC.

An iodine-125 radiolabelled M75, CA IX-selective mAb, has been developed for pre-clinical imaging of CA IX in hypoxic tumors in mouse xenograft models (*Chrastina, Pastoreková & Pastorek, 2003*; *Chrastina et al., 2003*). In addition, human A3 and CC7 mini-antibodies have been designed. Their small size enables them to distribute faster compared to full sized antibodies. These antibodies do not inhibit the catalytic activity of CA IX and are selective for the extracellular domain of human CA IX (*Ahlskog et al., 2009b*). By using mAbs coated with near-infrared fluorescent (NIRF) molecules, molecular imaging probes have been developed and applied for the non-invasive detection of breast cancer axillary lymph node (ALN) metastases. The high selectivity of these probes have been confirmed *in vitro* and *in vivo* using models of preclinical breast cancer metastasis (*Tafreshi et al., 2012*).

## Affibody molecules for imaging of CA IX expression

The affibodies are specially engineered small proteins that can bind to target proteins with a high affinity similarly to mAbs. These molecules can be used as novel anticancer drugs and/or for radionuclide imaging of tumors. In a recent study, several *in vitro* and *in vivo* properties of affibodies labeled with $^{99m}$Tc and $^{125}$I were characterized. Tested affibodies were highly specific to CA IX in SK-RC-52 cells and selectively accumulated in SK-RC-52 xenografts (*Garousi et al., 2016*). The study suggests the usefulness of CA IX-binding affibodies for cancer detection and therapy.

## Imaging of CA IX expression with small molecular chemical probes

Chemical probes can be applied for labeling and detection of biomolecules in order to study molecular processes occurring within living cells. The sulfonamide-based CA inhibitors efficiently bind to CA IX in hypoxic tumors as the active site of the enzymes is only available upon hypoxic conditions (*Svastová et al., 2004*). Unlike CA IX-specific mAbs, sulfonamides can recognize cells that are in hypoxic conditions. Thus, CA IX inhibitors and mAbs can give the different information about imaging and prognosis (*Pastorekova, Ratcliffe & Pastorek, 2008*). To prevent the sulfonamide-based inhibitors from passing through the membrane, inhibitors can be conjugated with fluorescent dye (FITC), albumin or hydrophilic sugar moieties that would prevent their entry into the cell (*Li et al., 2009b*). Among them, sulfonamides attached to FITC were shown to be membrane-impermeable with a high affinity to CA IX. This imaging agent was able to bind

to CA IX, expressed in cells under hypoxic but not normoxic conditions (*Svastová et al., 2004*). Similarly, acetazolamide-based derivatives bearing many types of NIRF dyes were designed as promising probes for the imaging of hypoxia-induced CA IX in tumor cells. Compounds were characterized to be up to 50-fold selective to CA IX compared to CA II. In preclinical studies using mice with HT-29 tumors, the significant impact of CA IX inhibitors with NIRF group on the non-invasive quantification of CA IX was determined (*Groves et al., 2012*). Moreover, fluorescent sulfonamides containing a charged fluorophore have been used *in vivo* and have shown a great efficiency in detecting CA IX in HT-29 and SK-RC-52 tumor xenografts (*Cecchi et al., 2005*; *Ahlskog et al., 2009a*).

## Imaging hypoxic tumor areas with nonpeptidic ligand conjugates

Recently, nonpeptidic ligand conjugates have been evaluated for single-photon emission computed tomography (SPECT) imaging of hypoxic cancers that express CA IX (*Lv, Putt & Low, 2016*). For a better clinical care, a broader knowledge about the level of hypoxia is needed. CA IX-targeting ligand was synthesized with the aim to deliver the attached $^{99m}$Tc-chelating agent to hypoxic regions. The studies of binding characterization *in vitro* and imaging of the biodistribution *in vivo* were carried out. Results showed that several such conjugates can selectively bind to CA IX in tumors. This study revealed the significantly important applications of nonpeptidic ligand conjugates to evaluate the level of hypoxia in tumors (*Lv, Putt & Low, 2016*).

In summary, the mAbs G250 and M75 have the advantages of binding to CA IX selectively on the surface of cancer cells, and thus they are able to detect cancer cells that overexpress CA IX. This is because the mAbs are raised against specific epitopes of CA IX, and they are unable to pass through the cell membranes due to the high molecular weight. However, the mAbs (G250 and M75) bind to the PG domain, and therefore they cannot affect its catalytic activity. In contrast, chemical inhibitors recognize the active site and can inhibit the enzymatic activity of CA IX, but they might possess several disadvantages including the low selectivity because of similarity of the α-CAs active sites, and the permeation through the plasma membrane. Thus, they might have off-target effects because of affinity to both intracellular and extracellular CAs. If the chemical inhibitors are conjugated with bulky molecules to avoid the internalization, they may still bind to other membrane CAs, such as CA XII. Thus, the properties of mAbs and chemical inhibitors need to be taken into consideration for using them as anticancer agents or as probes for the imaging of solid tumors.

## CONCLUSION

The critical role of CA IX in the tumor progression and aggressiveness has been shown and CA IX has been proposed as a promising therapeutic drug target and a clinically useful biomarker of the broad range of hypoxic tumors. Our review described efforts in the development of selective agents against CA IX. It is a challenging task to develop a compound of high affinity and selectivity towards only one CA isoform due to the high homology between twelve catalytically active CA isoforms in human body. Deeper insight in the structural analysis and interactions of proteins involved in pH regulatory mechanisms

of tumor cell could provide the relevant new strategies for rational drug design of CA IX-selective compounds for the therapy and diagnostic imaging.

**List of abbreviations**

| | |
|---|---|
| **ADCs** | Antibody-drug conjugates |
| **ALN** | Axillary lymph node |
| **EZA** | Ethoxzolamide |
| **IC$_{50}$** | The concentration causing 50% inhibition of target activity |
| **CA** | Carbonic anhydrase |
| **ccRCC** | Clear cell renal cell carcinoma |
| **FITC** | Fluorescent dye |
| **FTSA** | Fluorescent thermal shift assay |
| **HIF** | Hypoxia-inducible factor |
| **ITC** | Isothermal titration calorimetry |
| **mAbs** | Monoclonal antibodies |
| **NIRF** | Near-infrared fluorescent |
| **PET/CT** | Positron emission tomography/computed tomography |
| **SFA** | Stopped-flow $CO_2$ hydration assay |
| **SLC** | SignalChem Lifesciences Corp |
| **SMDCs** | Small molecule-drug conjugates |
| **SPECT** | Single-photon emission computed tomography |
| **SPR** | Surface plasmon resonance |
| **U-104** | Ureidosulfonamide inhibitor of CA IX |

### Funding

The work was supported by grants from the Research Council of Lithuania (Daumantas Matulis, grant number S-MIP-17-87), Jane & Aatos Erkko Foundation (Seppo Parkkila), Sigrid Jusélius foundation (Seppo Parkkila), and Academy of Finland (Sepo Parkkila). The funders had no role in study design, data collection and analysis, decision to publish, or preparation of the manuscript.

### Grant Disclosures

The following grant information was disclosed by the authors:
Research Council of Lithuania: S-MIP-17-87.
Jane & Aatos Erkko Foundation.
Sigrid Jusélius foundation.
Academy of Finland.

### Competing Interests

Daumantas Matulis declares that he has patents and patent applications for CA inhibiting compounds. Ashok Aspatwar and Seppo Parkkila are currently employed by Fimlab Ltd. Other authors confirm that this article content has no conflicts of interest.

## Author Contributions

- Justina Kazokaitė and Ashok Aspatwar conceived and designed the experiments, performed the experiments, analyzed the data, wrote the paper, prepared figures and/or tables, reviewed drafts of the paper.
- Seppo Parkkila and Daumantas Matulis conceived and designed the experiments, analyzed the data, wrote the paper, reviewed drafts of the paper.

## Data Availability

This is a review article that did not generate any additional previously unpublished raw data.

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
