# Peer review of "An update on anticancer drug development and delivery targeting carbonic anhydrase IX"

_PeerJ, doi:10.7717/peerj.4068_

## Round 0.1 · original submission · Major Revisions

The authors should carefully read the comments of all the four referees. Particularly I would emphasize that a review of literature should be comprehensive and it should highlight all the work in literature and not be a simple self-satisfaction.

Reviewer 1 ·

Basic reporting

The title of the review article is not reflecting at all the content of the work, since only sulfonamides are trated, and more precisely the few sulfonamides described by Matulis group. Although the authors mentioned the five types of small molecule CA inhiibtors targeting thismisoform, then only a very limited number of compounds is considered.

Experimental design

There are no experiments presented. The various methods of inhibitor assay are not very meaningful in a revie article on such a specialized area

Validity of the findings

Two structures of CA IX inhiibtors are presented which are of little interest since such compounds did not ptogress even to preclinical work. The part on antibodies is very badly written and antibody-drug conjugates not even mentioned

Additional comments

Overall this is a badly written ms which has a mixture of diverse topics whoch do not for, a solid ms, being too heterogeneous. For example it is difficult to understand what the zebra fish model did for the discovery of CA IX inhibitors.

Reviewer 2 ·

Basic reporting

The language is clear, but the selection of the articles for this review is absolutely authors-oriented!

Experimental design

not applicable

Validity of the findings

Table 1: I could not understand or quantitate the differences between "minor effect on pH" and "significant effect on pH".
is CA IX expressed in Xenopus transmembrane (as in humans) or cytosolic? This could be detrimental for the significance of these data. And maybe this part, dealing with only two CA inhibitors could be deleted because it is not validated for this target in the literature.

Reviewer 3 ·

Basic reporting

See comments to the authors

Experimental design

See comments to the authors

Validity of the findings

See comments to the authors

Additional comments

This review was written by an expert team who has developed new molecules to target carbonic anhydrase IX and has contributed significantly to the field. However, I think it is highly focussed on their own work rather than a general review, for example, only focussing on one subclass of inhibitors, the aromatic sulfonamides, so it is more of a report of their work only, where they, for example, comment “for the first time we emphasised the use of mammalian models”. However, they have already published this in 2016 and 2017, so anyone looking at the literature would find this.

They also make several claims, for example, “the first review to combine information about animal models and other methods used in the field”, whereas there are many reviews on carbonic anhydrase IX as a target. In fact, it would be useful to include other recent reviews in the references. Essentially, it is partly a technical description of work they have done in terms of the sorts of assays that have been around or published already. They also cover the zebra fish work already published and is well recognised as a useful phenotypic screen.

CA-targeted strategies focused mainly on preclinical work and the clinical studies are poorly reported. For example, the carbonic anhydrase IX inhibitor in clinical study was indisulam, yet it has been discontinued from clinical study a couple of years ago now. Similarly, the Willex antibody was actually effective at predefined set points of expression of CA9, yet has not been commercially developed. Finally, SLC0111 has gone through phase I but there is no information on the pharmacodynamics used, toxicity, or the dosing or any other details. So for someone who is interested in the clinical development, there is out of date information, not enough discussion as to why apparently successful compounds are counted as failures or update on the most recent compounds or how they are proven or not to have an effect on carbonic anhydrase IX.

Overall, therefore, I felt it is mainly concerned with their own publications in the last two years and summarising those data without an adequate discussion of the clinical problems and why with over 1000 compounds reported by Supura, onlya single compound has progressed to phase II studies in over 15 years since it was recognised an important therapeutic target. A much more detailed discussion of these problems would be of more interest.

·

Basic reporting

Kazokaité et al. have submitted a review article summarazing the current status of development of anti-tumor agents targeting carbonic anhydrase IX (CA IX). CA IX acts in control of tumor pH, modulates cell adhesion and invasion and contributes to tumor progression. Therefore, CA IX is ranked among promising anti-tumor drug targets with many new developed CA IX-targeting molecules every year. For that reason the topic of submitted review is actual and beneficial.
Professional English is used throughout the review, but in some parts is the text more complex than necessary. The number of references is fully sufficient for this type of a review, some of the references that I would recommend to add will be mentioned later There are no figures or charts present in publication, I would suggest adding at least one or two charts to improve readability. The review contains two helpful tables, but Table 2 has incorrect text wrapping in several columns.

Experimental design

'no comment'

Validity of the findings

'no comment'

Additional comments

1. Line 55: „The unregulated pH is a common feature of many aggressive tumors to adapt to hypoxia.“ I lack the clarification of the relationship between hypoxia and the problem with pH regulation. Explanation of the extracellular acidification of tumor microenvironment is given below, but there is no link to hypoxia.
2. In text, the terms „antitumor“ „anti-tumor“ „anticancer“ „anti-cancer“ are used extensively. It is necessary to unify the writing of this term, with or without a dash.
3. I suggest that the text from lines 157-166 is not thematic in the chapter. The division of inhibitors according to CA inhibition mechanisms is not important in describing the SFA method.
4. Lines 240-242: I wonder what resulted from discovery of a compound that is so tight bound to CA IX that the binding can not be measured by the SPR method? Is such compound a good candidate for CA IX-targeting agent? And is the SPR method appropriate for searching for suitable ligands?
5. Lines 291 and 297: It is necessary to add the number of carbonic anhydrase IX to headings. There is only „CA-targeted“ but the meaning is „CA IX-targeted“.
6. Line 305: I think that in the phrase "who are at risk of relapse" it would be appropriate to add „who are at high risk of relapse“, because unfortunately there is a risk of relapse for all patients.
7. Capture 4.1: In this part you do not mention the monoclonal antibody VII/20, which also has the therapeutic potential. I would recommend adding data from the publications Zatovicova et al. 2003 and Zatovicova et al. 2010.
8. Line 340: I would recommend changing the word order in the sentence to: „To demonstrate the anti-tumor effect of CA IX inhibition in vivo,...“
9. Line 359: There is no specification which „compounds may be suitable...“
10. Line 430: The use of the term "hypoxic cancers" does not seem to me to be appropriate. In my opinion, it would be better to use, for example, "hypoxic tumor areas".
11. Line 442: You note that antibodies binding does not affect the catalytic activity of CA IX. Could it be possible that it is affected by antibodies with the ability to internalize CA IX protein?

---

## Round 0.2 · accepted · Accept

The paper has been amended accordingly to referee's comments.

Reviewer 2 ·

Basic reporting

the manuscript is now suitable for publication

Experimental design

not applicable

Validity of the findings

suitable

Additional comments

none